# Relationship of prostate cancer topography and tumour conspicuity on multiparametric magnetic resonance imaging: a protocol for a systematic review and meta-analysis

Pranav Satish ,[1] Alex Freeman,[2] Daniel Kelly ,[3] Alex Kirkham,[4] Clement Orczyk,[5] Benjamin S Simpson ,[6] Francesco Giganti,[4] Hayley C Whitaker,[7] Mark Emberton,[5,7] Joseph M Norris [7]

For numbered affiliations see end of article.

**Correspondence to**
Joseph M Norris;
joseph.norris@ucl.ac.uk

## ABSTRACT

**Introduction** Multiparametric magnetic resonance imaging (mpMRI) has improved the triage of men with suspected prostate cancer, through precision prebiopsy identification of clinically significant disease. While multiple important characteristics, including tumour grade and size have been shown to affect conspicuity on mpMRI, tumour location and association with mpMRI visibility is an underexplored facet of this field. Therefore, the objective of this systematic review and meta-analysis is to collate the extant evidence comparing MRI performance between different locations within the prostate in men with existing or suspected prostate cancer. This review will help clarify mechanisms that underpin whether a tumour is visible, and the prognostic implications of our findings.

**Methods and analysis** The databases MEDLINE, PubMed, Embase and Cochrane will be systematically searched for relevant studies. Eligible studies will be full-text English-language articles that examine the effect of zonal location on mpMRI conspicuity. Two reviewers will perform study selection, data extraction and quality assessment. A third reviewer will be involved if consensus is not achieved. The Preferred Reporting Items for Systematic Reviews and Meta-Analyses guidelines will inform the methodology and reporting of the review. Study bias will be assessed using a modified Newcastle-Ottawa scale. A thematic approach will be used to synthesise key location-based factors associated with mpMRI conspicuity. A meta-analysis will be conducted to form a pooled value of the sensitivity and specificity of mpMRI at different tumour locations.

**Ethics and dissemination** Ethical approval is not required as it is a protocol for a systematic review. Findings will be disseminated through peer-reviewed publications and conference presentations.

**PROSPERO registration number** CRD42021228087.

## Strengths and limitations of this study

► This systematic review protocol follows the Preferred Reporting Items for Systematic Review and Meta-Analysis Protocols guidelines.
► This review addresses a gap in the literature with a detailed assessment of the crucial questions surrounding invisible multiparametric magnetic resonance imaging lesions.
► Due to the limited evidence on the topic, the included evidence will be retrospective cohort studies, limiting the strength of conclusions drawn for the thematic synthesis and meta-analysis.
► The degree of heterogeneity may limit the generalisability of the results.

## BACKGROUND

Multiparametric magnetic resonance imaging (mpMRI) has improved the triage of men with suspected prostate cancer, through precision prebiopsy identification of significant disease.

Nevertheless, anywhere from 7% to 55% of prostate cancer lesions may be overlooked by mpMRI.[1 2] As a result, identifying factors that influence visibility of clinically significant cancer has been the subject of extensive research over the past few years.[1 3 4] While multiple important characteristics, including tumour grade and size have been shown to affect conspicuity on mpMRI,[4 5] tumour location and association with mpMRI visibility remains an underexplored facet of this field.

Although the link between tumour location and conspicuity on mpMRI has not yet been fully elucidated, location does appear to be associated with clinical risk. Transitional zone (TZ) tumours have been associated with lower pathological grade disease[6] and lower rates of biochemical failure following radical prostatectomy.[7] There is also increasing evidence to suggest that mpMRI-visible lesions may associate with more aggressive clinical features.[8] Given this growing body of evidence linking risk, location and conspicuity on mpMRI, the

need to amalgamate these features is evident. Therefore, the objective of this systematic review and meta-analysis is to collate the extant evidence regarding the effect of prostate cancer zonal location on mpMRI conspicuity, for the first time. This review will help clarify mechanisms that underpin whether or not a tumour is visible, and the prognostic implications of our findings.

## METHODS AND ANALYSIS

This systematic review protocol has been written in line with the Preferred Reporting Items for Systematic Reviews and Meta-Analysis Protocol (PRISMA-P) 2015 checklist.[9] The review was prospectively registered with the PROS-PERO review database (CRD42021228087) and will be conducted according to the A MeaSurement Tool to Assess systematic Reviews 2 (AMSTAR 2) critical appraisal tool.[10] All methods described were established prior to the conduct of the review. A systematic review of the PubMed, MEDLINE, Embase and Cochrane databases will be conducted, with the search strategy containing medical subject heading (MeSH) terms for maximum yield of the relevant literature. The search will include MeSH terms for 'prostate cancer' and 'mpMRI,' combined via Boolean operators with synonyms for 'conspicuity' and 'location.' Articles identified by the search strategy will be uploaded to Rayyan,[11] a systematic review tool to facilitate the screening process. The included articles' reference section will be screened for further relevant literature not picked up by the initial search strategy.

## STUDY SELECTION

The screening process will be conducted by two independent reviewers, removing irrelevant articles based on titles and abstracts. Pertinent studies will be downloaded and their full text assessed to see if they meet the inclusion criteria. Any dispute between reviewers over the relevance of a study will be discussed until concordance is achieved, or a third reviewer will be consulted. The reasons for exclusion will be documented and detailed in the PRISMA flow diagram. Before starting screening, to minimise inter-reviewer bias, we will conduct calibration exercises to maintain consistency between the two reviewers.

## INCLUSION CRITERIA

Studies should compare MRI performance between different locations of the prostate in men with existing or suspected prostate cancer. Zonal location may be defined as traditional McNeal zones,[12] biopsy-derived data such as Barzell zones,[13] or data acquired via the Ginsburg protocol[14] or any other recognised approach of describing prostate topography. More general definitions of location such as anterior, posterior, base, midgland and apex will also be included. Visibility on mpMRI must be measured by any version of the Prostate Imaging-Reporting and

| Table 1 | Data extraction items | |
|---------|----------------------|-----------|
| **Item** | **Data extracted** | **Data type** |
| 1 | Author | Study characteristic |
| 2 | Publication year | Study characteristic |
| 3 | Study design | Study characteristic |
| 4 | Patient population | Demographic |
| 5 | Number of patients | Demographic |
| 6 | mpMRI protocol | Methodology |
| 7 | mpMRI scoring scheme | Methodology |
| 8 | Method of pathological correlation | Methodology |
| 9 | Definition of clinically significant cancer | Methodology |
| 10 | Definition of lesion visibility | Methodology |
| 11 | Zones assessed | Outcome |
| 12 | Differential quantification of conspicuity | Outcome |

mpMRI, multiparametric magnetic resonance imaging.

Data System (PI-RADS)[15] or Likert scoring system,[16] which is more commonly used in UK-based studies. Other mpMRI scoring schemes (eg, centre-specific approaches) will not be considered.

## EXCLUSION CRITERIA

Conference abstracts, expert opinions, correspondence articles and case reports will be excluded. Studies that do not correlate zonal location and mpMRI conspicuity will be excluded, as will studies not in the English-language.

## DATA EXTRACTION

Like the study selection process, reviewers will then independently extract the relevant study information using the Cochrane data extraction form. Discrepancies between the data extraction will be resolved by consensus. The following study characteristics will be collated: year of publication, authors, study design, patient population, number and age of study participants, mpMRI scoring scheme used (PI-RADS or Likert), definition of clinically significant disease, definition of tumour visibility, sample processing approach, zone(s) assessed and the differential quantification of conspicuity. Table 1 summarises data extraction items.

## ENDPOINTS

The primary endpoint will be the differential tumour visibility on mpMRI, as stratified by location. Secondary endpoints will include explanatory links between location

and mpMRI conspicuity, as well as the potential clinical implications. As study methodology may impact the determination of the zone, we will include these variables in a moderator analysis (see Meta-analysis section).

## ASSESSMENT OF BIAS

The Quality Assessment of Diagnostic Accuracy Studies 2 (QUADAS-2) tool to assess bias in diagnostic accuracy studies will be calculated across the included studies.[17] The tool assesses risk of bias across four domains: patient selection, index test(s), reference standard, flow and timing. Within each section, questions evaluate the quality of the research methodology, at the individual study level, outputting the binary outcome of 'high' or 'low' risk of bias. Both reviewers will independently calculate the score for each study, and any disagreement will be settled by consensus or the involvement of a third reviewer if consensus cannot be achieved. The outcome of the bias assessment will inform the thematic synthesis by providing an assessment of the reliability and applicability of the available evidence. If studies are deemed to be of excessively low quality (or high bias), then these may be excluded from the thematic synthesis at the discretion and consensus of both reviewers. Or, if included, will be accompanied by qualifying commentary in the discussion. For the meta-analysis, studies with 'high' risk of bias in two or more domains will be excluded.

## META-ANALYSIS

As previously described,[18] if there are a sufficient number of studies available that analyse location, using sufficiently comparable methodologies, then we will conduct a meta-analysis. The purpose of the meta-analysis is to obtain a single pooled effect for MRI performance in imaging different areas of the prostate. To limit the effect of methodological heterogeneity on the pooled diagnostic accuracy estimate, we will use two criteria for inclusion in the meta-analysis. First, all studies in the meta-analysis must use mpMRI, rather than single sequence imaging. Second, they should define a positive MRI as a PI-RADS/Likert score of ≥3. Finally, effect estimates for non-randomised and randomised studies will be pooled in separate meta-analyses, as the non-randomised studies will be more affected by bias.

For each topological location reported across studies, the total number of lesions with prostate cancer ($m^i$), the total number of patients without prostate cancer ($n^i$), the true positives ($y^i$, mpMRI visible lesions with confirmed cancer), false positives ($z^i$, mpMRI visible lesions which are histologically negative), true negatives ($m^i - y^i$, mpMRI invisible and histologically negative) and false negative ($n^i - z^i$, mpMRI invisible and present on histological examination) will be recorded. In the case of studies which do not give sufficient detail to derive these values but do list sensitivity or specificity and total patient numbers, these values will be reverse engineered, for example in

the case of a study which reports mpMRI sensitivity for a TZ lesion at 0.788 in a group of 60, the true positives can be calculated as $y^i = m^i p^i$ rounded to the nearest integer or $TP = 60 \times 0.788 = 47.28$ (47.3 true positives). The respective sensitivities and specificities of mpMRI for tumours within differing zonal locations will then be compared. The distribution of untransformed, logit and double arcsine transformed sensitivities and specificities will be compared. Whichever distributions resemble a normal distribution (assessed using density plots and Shapiro-Wilk tests) will be used for further analysis.

The model fitted will be determined based on inter-study variation (measured via $I^2$); if significant, a random-effect model will be fitted, or a fixed-effect model if not. After fitting a model to all relevant studies, leave-one-out (LOO) analyses and accompanying diagnostic plots will be used to identify influential studies, including externally studentised residuals, difference in fits values, Cook's distances, covariance ratios, LOO estimates of the amount of heterogeneity, LOO values of the test statistics for heterogeneity, hat values and weights. Studies with a statistically significant influence on the fitted model will be removed as outliers and the model refitted. These outliers will be examined for potential confounding variables such as study methodology or poor interobserver agreement for mpMRI scans. Predicted sources of heterogeneity in mpMRI performance include interstudy variation in MRI reader experience, the MRI scoring system used and the MRI magnet strength used, so these will be accounted for in the interpretation of the results. For each tumour location, individual models will first be fit and the pooled sensitivities, specificities, negative predictive value and positive predictive value will be reported. Finally, using tumour location as a subgroup, a bivariate model will be fit to the data and any differences in sensitivity and specificity will be compared using SROC curves and tested using the $\chi^2$ likelihood ratio test.

If appropriate, moderator analysis will also be performed between factors, which may influence the outcome, including, but not limited to, study methodology, study size, year of publication, cohort type, relevant methodology (such as radical prostatectomy vs biopsy studies) or visibility definition. Analytics will be performed as outlined by Wang.[18] If there is insufficient data to conduct a meta-analysis, only thematic synthesis will be performed.

## DISCUSSION

Following Level 1 evidence support of mpMRI-directed prostate cancer diagnosis[1 19–21] and implementation into national and international clinical guidelines,[22] mpMRI has moved to the forefront of prostate cancer diagnosis. While mpMRI provides refinement of the traditional transrectal ultrasound-guided biopsy approach,[1] not all significant prostate cancers are visible on mpMRI, and it is pertinent for us now to explore the reasons that underpin this inconspicuity. Through this systematic review, we

will identify the correlation between zonal location and conspicuity, to enhance our understanding of the factors that influence prostate cancer visibility on mpMRI.

Unusual radiological manifestations of prostate cancer are not uncommon due to nuanced histopathological landscape in each zone, rendering MRI signal generation highly varied. In the peripheral zone, several radiological 'hiding places' exist, such as very distal apical tumours, subcapsular crescentic tumours, tumours mimicking the posterior midline and infiltrative peripheral zone tumours.[23] In the anterior prostate, the presence of benign prostatic hyperplasia (BPH) in the TZ has been known to generate false-positive results, due to difficulty discriminating prostate cancer from BPH nodules.[24] Furthermore, MRI signal properties of TZ tumours and normal TZ tissue can be similar,[25] further complicating this issue. In the apex of the prostate, the apical capsule is less conspicuous on imaging, and periprostatic fat is usually sparse or completely absent in the apex, which may make the tumour less visible.[26] Nevertheless, one study showed high detection rates for this type of tumour.[27] Given the many, sometimes contradictory, stances surrounding zonal location and conspicuity in the literature, the planned systematic review will discuss these features, among others, in detail.

In summary, this systematic review and meta-analysis will combine the extant evidence in this emerging field, for the first time. Collation and analysis of these data will enrich our understanding of the effect of zonal location on the conspicuity of prostate cancer on mpMRI. Additionally, this process will help reveal the potential clinical role that these effects play in both diagnosis and treatment, thereby aiding identification of important avenues for future research.

## TRIAL STATUS

1. Preliminary searches: started.
2. Piloting of the study selection process: started.
3. Formal screening: started
4. Data extraction: not started.
5. Risk of bias assessment: not started.
6. Data analysis: not started.

## Draft of search strategy for MEDLINE, Embase, PubMed and Cochrane databases

(Prostate cancer OR prostate tumo*r OR prostate lesion)

AND (region* OR position* OR location* OR zone OR zonal location OR midgland OR base OR apex OR AFMS OR anterior OR posterior)

AND (mpMRI OR multiparametric MRI OR multiparametric MR imaging OR mp-mri)

AND (conspicuity OR visibility OR detect* OR visuali* OR identif*)

## ETHICS AND DISSEMINATION

Due to the nature of the study, there are no relevant ethical concerns and informed consent will not be required. The protocol and systematic review will be disseminated via a peer-reviewed journal.

**Author affiliations**
[1]UCL Division of Surgery & Interventional Science, UCL Medical School, London, UK
[2]Department of Pathology, University College London Hospitals NHS Foundation Trust, London, UK
[3]School of Healthcare Sciences, Cardiff University, Cardiff, UK
[4]Department of Radiology, University College London Hospitals NHS Foundation Trust, London, UK
[5]Department of Urology, University College London Hospitals NHS Foundation Trust, London, UK
[6]UCL Cancer Institute, University College London, London, UK
[7]Division of Surgery and Interventional Science, University College London, London, UK

**Contributors** PS and JMN drafted the manuscript and created the study concept. BSSS provided the meta-analysis section. FG, AF, DK, AK, CO, HCW and ME provided supervision and guidance. All authors reviewed and approved the manuscript in its current form.

**Funding** JMN is funded by the Medical Research Council (MRC) on an MRC Clinical Research Training Fellowship (MRC Grant Ref: MR/S00680X/1). BSS receives funding from the Rosetrees Foundation.

**Competing interests** JMN received funding from the MRC. BSSS received funding from the Rosetrees Trust. HCW received funding from PCUK, the Urology Foundation and the Rosetrees Trust. AK, AF and ME have stock interest in Nuada Medical. ME received funding from NIHR-i4i, MRC, Sonacare, Trod Medical, Cancer Vaccine Institute and Sophiris Biocorp for trials in prostate cancer. ME is a medical consultant to Sonacare, Sophiris Biocorp, Steba Biotech, GSK, Exact Imaging and Profound Medical. Travel allowance was previously provided from Sanofi Aventis, Astellas, GSK and Sonacare. ME is a proctor for HIFU with Sonacare Inc and paid for training other surgeons in this procedure.

**Patient and public involvement** Patients and/or the public were not involved in the design, or conduct, or reporting or dissemination plans of this research.

**Patient consent for publication** Not applicable.

**Provenance and peer review** Not commissioned; externally peer reviewed.

**ORCID iDs**
Pranav Satish http://orcid.org/0000-0003-4531-9114
Daniel Kelly http://orcid.org/0000-0002-1847-0655
Benjamin S Simpson http://orcid.org/0000-0003-3685-6110
Joseph M Norris http://orcid.org/0000-0003-2294-0303

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
