## [Reviewer comments · BMJ Open]

ARTICLE DETAILS

TITLE (PROVISIONAL)	Relationship of prostate cancer topography and tumour conspicuity on multiparametric magnetic resonance imaging: a protocol for a systematic review and meta-analysis
AUTHORS	Satish, Pranav; Freeman, Alex; Kelly, Daniel; Kirkham, Alex; Orczyk, Clement; Simpson, Benjamin Scott; Giganti, Francesco; Whitaker, Hayley; Emberton, Mark; Norris, Joseph

VERSION 1 – REVIEW

REVIEWER	Carlsson, Sigrid Memorial Sloan-Kettering Cancer Center
REVIEW RETURNED	31-Mar-2021

GENERAL COMMENTS	Thank you for the opportunity to review this manuscript, which describes the protocol for a planned systematic review and meta-analysis (SR-MA). The manuscript is well-written and the methodology is sound. The authors follow the PRISMA guideline, provide detailed methodology for their planned SR-MA and have a solid plan to assess both risk of bias and heterogeneity. The protocol is pre-registered in PROSPERO, which is terrific. Attention to the following points could improve the quality of the reporting in the protocol: 1. The reviewer recommends that the authors carefully review the AMSTAR-2 checklist (A Measurement Tool to Assess systematic Review) (available at https://amstar.ca/Amstar_Checklist.php + guidance document https://amstar.ca/docs/AMSTAR%202-Guidance-document.pdf). This will ensure that, upon completion, the SR-MA will be judged to be of high-quality at the time of peer-review of the final submitted manuscript. Because this manuscript describes the study protocol for this SR-MA, the authors now have a golden opportunity to design their SR with all the AMSTAR details in mind and embark on a careful and comprehensive review process with the goal of producing a high-quality end-product. For instance: a) Can the PICO be better described? As currently written in the abstract, the objective is: “to collate the extant evidence regarding the effect of prostate cancer zonal location on mpMRI conspicuity”. This might be a bit unclear for an uninitiated reader of the future target journal. (AMSTAR-2 item 1)b) Can the exact review criteria be described in a bit more detail? (AMSTAR-2 item 2)c) Please describe the process for study selection and data extraction by the two independent reviewers in more detail. (AMSTAR-2 items 5-6)
--

	d) Will both RCTs and observational studies be included? Please justify. If yes, how will they be synthesized quantitatively? If only observational studies, why? Please justify (several AMSTAR-2 items) e) In addition to the very nice and detailed section about statistical methods to examine heterogeneity and outliers, what is the plan to investigate the SOURCES of heterogeneity to try to determine the factors that might lead to differences in study results? (several AMSTAR-2 items) For example, different mpMRI imaging protocols, different approaches to determining zonal location, different readers etc. What exactly do the authors plan to look for when examining heterogeneity? These are just some examples and if the authors review the checklist carefully and revise the protocol manuscript with these items in mind, it will ensure a high-quality end product. 2. In addition to the terrific list of items in Table 1, will the authors be using any other data extraction tool? For example, Cochrane has such tools (https://dplp.cochrane.org/data-extraction-forms). 3. What is meant by “a thematic approach will be used to synthesise key location-based factors associated with mpMRI conspicuity” and “only thematic synthesis”. Do the authors mean a qualitative synthesis using "themes" rather than a quantitative (meta-analysis)? What method will be used for summarizing articles in narrative form? 4. “If there are enough eligible studies, a meta-analysis will be conducted.” How many is “enough”? Two? More? How will this be determined? How many articles do the authors expect to find? Have the authors done a quick “pilot” literature search to get a sense of how many papers are currently out there? 5. What “gold standard” will be used to determine the true disease state for tumor lesion location in the included studies? Systematic biopsy (how many cores?), MRI-targeted biopsy, template mapping biopsy, saturation biopsy, RP pathology.....? How will this be accounted for in the MA?
--	---

REVIEWER	Rocco, Bernardo University of Modena and Reggio Emilia
REVIEW RETURNED	11-Apr-2021

GENERAL COMMENTS	The link between tumor location and conspicuity of PCa at mpMRI has not been fully explored, unlike tumor grade and size. Since “mpMRI has recently moved to the forefront of PCa diagnosis”, as correctly stated by Authors, insights about the relationship between tumor location and MRI findings devoid a further analysis. From a methodological point of view, the protocol seems adequate; from a personal urological point of view, outcomes may be of added value in determining the role of mpMRI in PCa diagnostic pathway. As a separate analysis (or as an added endpoint) I would suggest to consider also the relationship between mpMRI findings stratified by zonal anatomy and the occurrence of extraprostatic extension of cancer. It would suggest the surgeon to rely – or not – on
--

	mpMRI for surgical pre-planning taking into account also the location of cancer.
--	--

VERSION 1 – AUTHOR RESPONSE

Reviewer #1:

1a) We have attempted to detail the PICO more clearly by stipulating that ‘Studies should compare MRI performance between different locations of the prostate in men with existing or suspected prostate cancer’.

1b) Compliant with AMSTAR item 2, we included the review question, search strategy, inclusion/exclusion criteria, risk of bias assessment, meta-analysis plan and a plan for investigating causes of heterogeneity

1c) Compliant with AMSTAR items 5 and 6 we have added that both the study selection and data extraction process will be conducted in duplicate by two independent reviewers, with a consensus process in case of disagreement.

1d) We have added, with justification, that RCT and observational studies will be quantitatively synthesized separately. However, we suspect that there will be a dearth of randomised studies that examine location, as it is an underexplored facet of the prostate imaging field.

1e) To investigate sources heterogeneity, studies with outlier findings will be reassessed for potential causes of heterogeneity. These include, but are not limited to, variation in MRI reader experience, the MRI scoring system and the MRI magnet strength used. Any interstudy variation will be accounted for in the discussion.

2) We will use the Cochrane data extraction tool linked, thank you for the suggestion.

3) We have used the phrases ‘thematic approach’ and ‘thematic synthesis’ as interchangeable terms. We will summarise articles in a narrative form by grouping them according to recurring topography-related themes identified in the literature. These could include: biparametric vs multiparametric imaging, central zone imaging and apical prostate imaging.

4) While there is no formal threshold for the number of studies required for a meta-analysis, we have a semi-arbitrary minimum of three studies, as this would allow the identification of a study which is a clear outlier compared to the other two. This would facilitate a small meta-analysis. Pilot literature searches have indicated at least 10 studies with data on MRI performance between prostate zones. However, to quantitatively synthesis, we need access to the raw data, which is rarely reported in a usable way. As such, we will rely on correspondence with the authors to acquire data.

5) Various gold standards will be used in the thematic synthesis, to account for the diversity in the literature, including systematic biopsy (all cores), MRI-targeted, template mapping, saturation and RP specimens. The limitations of each will be discussed in the interpretation of the results and risk of bias assessment. In the quantitative synthesis, a separate analysis for radical prostatectomy specimens and biopsy will be conducted, to reflect the distinction between MRI performance at the diagnostic level versus men with confirmed cancer.

Reviewer #2:

Thank you for your suggestion. We agree, consideration of the occurrence of extraprostatic extension in regard to tumour location would prove useful during pre-surgical planning. However, we believe that, while important, occurrence of extraprostatic extension is outside the scope of this MRI diagnostic accuracy study.

VERSION 2 – REVIEW

REVIEWER	Carlsson, Sigrid Memorial Sloan-Kettering Cancer Center
REVIEW RETURNED	17-Sep-2021
GENERAL COMMENTS	The authors have done a nice job responding thoroughly to the reviewer's prior concerns and have revised their protocol accordingly. As a result, the quality has improved significantly. I have no further comments. Best wishes with your systematic review.